# Achievements in Thermosensitive Gelling Systems for Rectal Administration

**DOI:** 10.3390/ijms22115500

**Published:** 2021-05-23

**Authors:** Maria Bialik, Marzena Kuras, Marcin Sobczak, Ewa Oledzka

**Affiliations:** Department of Biomaterials Chemistry, Chair of Analytical Chemistry and Biomaterials, Faculty of Pharmacy, Medical University of Warsaw, 1 Banacha St., 02-097 Warsaw, Poland; maria.jurkitewicz@wum.edu.pl (M.B.); marzena.kuras@wum.edu.pl (M.K.); marcin.sobczak@wum.edu.pl (M.S.)

**Keywords:** rectal gel, thermosensitive gel, solid suppository, rectal drug delivery, drug delivery system, thermosensitive liquid suppository, poloxamer

## Abstract

Rectal drug delivery is an effective alternative to oral and parenteral treatments. This route allows for both local and systemic drug therapy. Traditional rectal dosage formulations have historically been used for localised treatments, including laxatives, hemorrhoid therapy and antipyretics. However, this form of drug dosage often feels alien and uncomfortable to a patient, encouraging refusal. The limitations of conventional solid suppositories can be overcome by creating a thermosensitive liquid suppository. Unfortunately, there are currently only a few studies describing their use in therapy. However, recent trends indicate an increase in the development of this modern therapeutic system. This review introduces a novel rectal drug delivery system with the goal of summarising recent developments in thermosensitive liquid suppositories for analgesic, anticancer, antiemetic, antihypertensive, psychiatric, antiallergic, anaesthetic, antimalarial drugs and insulin. The report also presents the impact of various types of components and their concentration on the properties of this rectal dosage form. Further research into such formulations is certainly needed in order to meet the high demand for modern, efficient rectal gelling systems. Continued research and development in this field would undoubtedly further reveal the hidden potential of rectal drug delivery systems.

## 1. Introduction

Oral administration is the preferred route for regular pharmacotherapy, as it is the easiest and most convenient. However, this is not feasible or even impossible in some cases (e.g., during nausea/vomiting or convulsions, in non-cooperative patients, prior to surgery). In such situations, the rectal route can be a viable option, and rectal administration is now well known for drug delivery. Traditionally solid suppositories are the most popular delivery systems applied for rectal drug administration and account for more than 98% of all rectal dosage forms. Traditional solid-type suppositories, however, often feel alien and uncomfortable to patients, encouraging refusal. To solve these problems, it would be desirable to develop a thermosensitive liquid suppository [1,2]. Such a formulation is administered to the anus simply, does not damage the mucosal layers, behaves as mucoadhesive to the rectal tissue without leakage and reduces the feeling of a foreign body. Thermosensitive liquid suppositories can be used for a variety of drugs, such as anticancer, analgesic, etc. Furthermore, thermosensitive systems enable superior control of drug release by varying the type and concentration of components. This solution may have a positive effect on pharmacotherapy effectiveness. The properties of this dosage form depend on the type of components used (thermosensitive and mucoadhesive polymers) and their concentration [3]. The purpose of this review is to summarise the concepts, applications and recent advances in thermosensitive liquid suppositories as Drug Delivery Systems (DDSs) of analgesic, anticancer, antiemetic, antihypertensive, psychiatric, antiallergic, anaesthetic and antimalarial drugs and insulin, with a primary emphasis on the type and concentration of their components, as well as on the rheological and pharmacokinetic/pharmacodynamic properties of this promising drug formulation.

### 1.1. Oral Route and Its Limitations

Many drugs are intended for oral use in the pharmaceutical industry. Oral drug delivery accounts for more than 50% of the global demand for drug delivery. The oral route (per oral (p.o.) is the most favoured route for drug administration, because it offers the highest degree of patient compliance. The benefits of oral administration are as follows:convenient—can be self-administered, pain-free, easy to take;absorption—occurs along the entire length of the gastrointestinal tract (GIT);cheap—compared to most other drug formulations [1,2].

For an oral drug, the time to onset of a desired pharmacological effect depends on a number of successive steps: dissolution of the formulation, passage to the site of absorption, permeation through physiological membranes, entry into the portal vein circulation, distribution from plasma to the site of action, and interaction with the receptor [4,5].

Oral administration is very effective for drugs with high solubility and gastrointestinal permeability. However, effective oral drug delivery with poor solubility and/or permeability and/or metabolic stability is quite challenging. In general, these drugs must be administered at a high dose to achieve therapeutic concentration [2]. Furthermore, due to their susceptibility to extreme GIT conditions and the risk of chemical or enzymatic degradation, many of them have a low percentage of absorption via the oral route. The pH of the GIT varies with the location. For example, the stomach has an acidic pH, but the pH in the intestine is in the range of 6.8–7.4. Drugs, such as artemether, erythromycin, candesartan cilexetil, undergo chemical degradation at acidic pH. For this reason, their bioavailability is much lower [2]. In addition, the various enzymes (esterases, lipases) present in the GIT result in the degradation of several drugs, such as antihyperlipidemic agents (simvastatin, ezetimibe) and cephalosporin antibiotics (cefpodoxime proxetil). Oral drug bioavailability, such as antidiabetic agents (repaglinide) and antihypertensive and cardiovascular agents (β-blockers, calcium channel blockers, angiotensin-converting-enzyme (ACE) inhibitors), is substantially low due to high levels of first-pass (hepatic) metabolism. Finally, drug efflux transporters such as P-glycoprotein are also responsible for the efflux of various drugs such as digoxin, paclitaxel, and doxorubicin from the site of absorption. It also reduces the bioavailability of the drugs [2,5,6,7,8].

In order to overcome these disadvantages, a number of formulation methods such as prodrugs (macromolecular conjugates), solid dispersions, self-microemulsifying drug delivery systems (SMEDDSs), microcapsules including liposomes, nanoparticles, P-glycoprotein inhibitor pre-treatment have been aimed to enhance drug physicochemical properties and to reach therapeutically relevant plasma drug concentrations [9,10,11,12,13]. However, the bioavailability of oral lipophilic medicines remains a major concern. Hence, there is a growing need to develop efficient DDSs to improve the bioavailability and therapeutic profile of a wide range of drugs. Among the many non-invasive routes available, the rectal route is a safe and promising alternative to drug delivery.

### 1.2. Rectal Route

Rectal dosage forms are the oldest pharmaceutical dosage forms seeing as their origin dates back to antiquity. Hippocrates mentions the different compositions of acorns that were rectal dosage types, and the Old Testament refers to “Magerarta”—a silver suppository. The first rectal dosage formulations consisted of solid supports (baked honey, soap, tallow, horn) impregnated with medicinal substances. These solid supports were replaced by cocoa butter at the end of the 18th century. The first mention of the addition of an active substance to the suppository mass was made by Henry and Guibourt in 1841 with the introduction of opium in cocoa butter [14].

In humans, the rectum is formed by the last 15 to 19 cm of the large intestine. The rectum has two or three curves within its lumen formed by submucosal folds [15]. The rectal wall is formed by an epithelium. Its luminal surface is covered by a membrane formed by 1 layer of cells, consisting of columnar epithelial, endocrine and goblet cells which secrete mucus. The surface area available for drug absorption in the rectum is approximately 200–400 cm^2^. The volume of fluid in the rectum is approximately 1–3 mL and is viscous. The rectal pH is constant and approximately 7.5–8.0; the temperature is usually 37 °C. The venous drainage of the rectum consists of three distinct veins. The upper haemorrhoid vein drains the upper rectum and empties into the haemorrhoidal vein, which flows into the liver. The middle and lower haemorrhoidal veins drain the rest of the rectum and return to the inferior vena cava (Figure 1). Absorption of water, sodium and chloride and secretion of potassium and bicarbonate occurs in the human colon, while active glucose and amino acid transport are lacking. Sodium and water absorption in the rectum is negligible [15,16,17,18].

The rectum has a relatively small absorption surface of 0.02–0.05 m^2^ and is characterized by a lack of villi. Drug absorption from rectal epithelium involves two transport routes: trans-cellular and para-cellular. The para-cellular route is the diffusion of drugs through space between epithelial cells, and the mechanism of uptake in the trans-cellular route depends on lipophilicity. Rectal drug absorption depends on several drug characteristics: small partition coefficient, large molecular size, charge and high capability of hydrogen bond formation are standard reasons for poor drug absorption [14]. The presence of faeces, which can also alter the absorption of the drug, is another hindrance. In the absence of faecal matter, the drug is more likely to come into contact with the absorbing rectal surface. Rectal content is usually alkaline; alkaline solutions are rapidly absorbed rather than acidic solutions. Other conditions, such as diarrhoea, colonic obstruction, and tissue dehydration, may influence the rate and degree of drug absorption from the rectal site. Aqueous and alcoholic solutions are absorbed rapidly, whereas suspensions and suppositories are absorbed slowly and constantly. The lower rectum is drained by the lower and middle haemorrhoidal veins and bypasses the liver, thereby avoiding, at least partially, the hepatic first-pass effect and enabling drugs to have systemic effects prior to metabolism in the liver. The rectal region is also massively drained by lymphatic circulation and may increase the systemic absorption of certain highly lipophilic drugs. Drug use through the rectal path is susceptible to both local and systemic drug delivery [14,19,20,21,22]. This route is used as an alternative to oral and invasive administration. Rectal drug delivery is essential if oral or intravenous treatment is not possible. Furthermore, it is ideally adapted for infants, children and geriatric patients [14,18,23,24].

The rectal route provides the following possible advantages for drug delivery over the oral route:rapid absorption of several drugs with low molecular weight;partial prevention of the first-pass metabolism, the potential for absorption into the lymphatic system;the retention of higher drug quantities;probability of efficient drug delivery and absorption [19,25,26];protection of enzymatically unstable drugs (insulin) owing to the lack of enzymes in the rectum [27,28];minimal first-pass drug metabolism given that the suppository is administered at an acceptable distance in the rectum [29];avoidance of exposure of gastric mucosa to irritant drugs such as non-steroidal anti-inflammatory drugs;increased effectiveness [30,31].

Conventional rectal dosage formulations are available in a number of countries. These forms have historically been used for localised treatments, including antipyretics, haemorrhoid and laxative therapy. Recent trends show an increase in the production of new rectal delivery systems to deliver drugs directly to the systemic circulation. In addition, higher bioavailability and controlled drugs release have become possible with the support of modern pharmaceutical products and innovative Rectal Drug Delivery Systems (RDDSs).

Such systems may be modified to act locally or systemically and may release the active agent immediately or over in an extended manner [16]. Traditional rectal dosage forms can be categorized into solid (suppository), liquid (enema, microenema, foam, suspension), semi-solid (gel, ointment) and medical devices (rectal tampon) [16]. Traditionally solid suppositories are the most common delivery systems used in rectal drug administration. They represent over 98% of all the rectal dosage forms. Suppositories containing a drug, solubilized or suspended in a suppository base, that upon melting or solubilization at physiological conditions, release the active substance for either local or systemic action [16,23,32]. Unfortunately, traditional solid suppositories suffer from disadvantages such as pain, an alien feeling and, thus, a patient’s refusal. A solid type suppository, which may reach the end of the colon, may also allow drugs carried to undergo the first-pass effect and many drugs are poorly or erratically absorbed across the rectal mucosa. It is also possible to metabolise drugs in microorganisms and rectal mucosa. In addition, certain suppositories are either “leaked” or expelled after insertion. In order to avoid these problems, the suppository should be inserted past the muscular sphincter, which is approximately half an inch in the rectum of infants and one inch in the rectum of children and adults [33,34]. In addition, the rectal absorption of most drugs is often erratic and unpredictable. The anus has a limited absorption surface area and may cause problems with dissolution due to the small fluid content of the rectum. In addition, in countries with a tropical climate, the proximity of its melting point to average room temperature is another drawback of conventional suppositories. Some of the problems in transport and storage could be observed. The optimal suppository should be easy to administer, without any pain during insertion, and should remain at the administered site to prevent the first-pass effect in the liver [19,23,35].

Traditional RDDSs have recently been improved by modifying the properties of the formulation, e.g., gelation temperature, gelation strength. Prolonged retention and controlled drugs release can help improve bioavailability and provide better pharmacokinetic profiles or local treatment effects. RDDS has been developed to provide greater control over the spread, retention and/or release of the drug through a range of formulation strategies. The types of novel RDDSs are as follows: hollow-type suppository, thermosensitive liquid suppository, mucoadhesive gel, micro and nanoparticles and vesicular drug delivery systems [16].

In particular, the problems of conventional solid suppositories related to the above can be solved by developing thermosensitive liquid suppositories. This material:forms a gel at a body temperature;has the required gel strength to prevent leaking out of the anus after administration;has sufficient bioadhesive force so as not to reach the end of the colon [16,22,36,37].

Furthermore, the simplicity of administration and the ability to reduce discomfort to the rectal mucosa has contributed to the development of these dosage forms [38,39].

### 1.3. Polymers and Their Properties Used in the Fabrication of Thermosensitive Liquid Suppositories

Liquid suppositories are thermosensitive rectal gels. They are also referred to as thermosensitive liquid suppositories due to the fact that the base material used in the formulation is a thermosensitive polymer that becomes a gel at physiological temperature (37 °C). The transition sol-gel temperature (T_sol-gel_) is below body temperature (<37 °C), making it possible to prepare liquid preparations, which gel back at body temperature [16,40]. Thermosensitive polymers such as poloxamers or pluronics in a proper concentration form a gel at physiological temperature, thus preventing leakage and excessive spreading in the rectum. In addition, the combination of mucoadhesive and thermosensitive polymers allows for more sustained drug release relative to the thermosensitive polymer alone [31,39,41,42]. Once in the rectal cavity, the development of mucoadhesive properties helps to immobilize the hydrogel for a prolonged period of time and to prolong the drug release, thereby favouring the systemic absorption of drugs [43]. Viscosity, gelling time and temperature threshold are all main elements in the preparation of thermosensitive liquid suppository [16,44,45].

The key benefits of thermosensitive liquid suppositories over traditional or solid suppositories are as follows:they are easy to administer to the anus as they remain liquid at lower temperatures,act as mucoadhesive to the rectal tissues preventing leakage after administration,do not cause any harm on mucosal layers,minimize the feeling of a foreign body compared with solid suppositories [3,16,41].

The most popular base of thermosensitive liquid suppository are poloxamers (triblock copolymers of poly(oxyethylene)-poly(oxypropylene)-poly(oxyethylene) (PEO–PPO–PEO)) [46]. They comprise a central block of hydrophobic polypropylene oxide (PPO) surrounded on both sides by the blocks of hydrophilic polyethylene oxide (PEO) [46] (Figure 2a). Triblocks copolymers (Poloxamer^®^ or Pluronic^®^ series) are the most commonly encountered thermosensitive systems in the pharmaceutical field. Mucoadhesive and thermosensitive polymers have gained much attention recently to decrease patient discomfort and relieve the alien feeling due to solid suppository insertion, and afford an endurable method of administration. Poloxamer solutions exhibit the phenomenon of reverse thermal gelation and remaining as solutions at low temperature (4 °C) and gelling again upon raising temperature (25–35 °C) [11]. In general, the phase transition temperature of poloxamers depends on their concentration. Poloxamer aqueous solutions stay fluid below T_sol-gel_, and the solution transforms to a semi-solid material above this temperature. Thermogelation is caused by hydrophobic interactions between the copolymer chains of poloxamer. When the temperature is raised, the poloxamer copolymer chains begin to aggregate into a micellar structure. The poloxamer in cold water acts as follows: the hydration layer surrounds the poloxamer molecule and the hydrophobic portion is separated due to hydrogen bonds. The hydrogen bonds break and the hydrophilic chains dissolve as the temperature increases. Poloxamer gelation is associated with a polymer dehydration process, which increases chain friction and entanglement while also producing hydrophobic association [3,38,47,48]. Poloxamers are also known for their compatibility with other compounds, high solubilization capacity of various active ingredients and good characteristics for active ingredient prolonged release [3,43,49,50].

Poloxamer 407 (P407, Pluronic F 127) is widely used because it allows the formation of colourless, transparent and easily washable water gels, which are non-irritating to the skin and mucous membranes. Its transition temperature is below body temperature, allowing for the preparation of liquid preparations that gel at body temperature. At ambient temperature, an aqueous solution of P407 with a concentration higher than 20% forms non-chemically cross-linked hydrogels. P407 solution has a lower phase transition temperature (<25 °C) at such a concentration, so solutions are gels at room temperature and thus difficult to use for drug delivery [51,52]. In order to create a temperature-responsive gel with a suitable phase transition temperature, poloxamer 188 (P188) is incorporated into P407 solutions to modulate the phase transition temperature [52,53]. The poloxamer mixture solutions have a higher transition temperature than the P407.

Despite many benefits, poloxamer hydrogels do not possess or have weak bioadhesive properties. In addition, they suffer a major disadvantage, which is low mechanical strength and high water permeability, limiting their use as a thermosensitive matrix [38]. If mucoadhesive properties are required, poloxamer must be formulated with other bioadhesive polymers. Other typical mucoadhesive polymers used in the preparation of liquid suppositories are: polyvinylpyrrolidone (PVP (Figure 2b)), sodium alginate, acrylic polymers (Polycarbophyl^®^ (PCP)), Carbopol^®^, and cellulose ether polymers such as carboxymethyl cellulose (CMC), hydroxypropyl methylcellulose (HPMC, Hypromellose), methylcellulose (MC, Metolose^®^ (Figure 2c)), hydroxyethyl cellulose (HEC) [43].

Thermosensitive polymers in aqueous solutions display temperature-dependent sol to gel transitions at specified levels. If the solidification appears above a certain temperature, the temperature is referred to as the lower critical solution temperature (LCST). Corresponding polymers are soluble below the LCST, even if solubility decreases above the LCST due to increased hydrophobicity. This contributes in a reversible gelation of the solution. In the drug delivery, the LCST should be between the average ambient temperature (25 °C) and the body temperature (37 °C). In this manner, the drug can be applied as a liquid and then transferred to gel after administration [54,55,56]. In the case of formulations showing sol-gel transition upon cooling one speaks of the upper critical solution temperature (UCST) [56,57,58]. Most natural polymers in aqueous solutions undergo sol-gel transformation at lower temperatures. However, some cellulose derivatives gel at higher temperatures. As the temperature increases, they gradually lose water, increasing intermolecular interactions, which are primarily mediated by methoxy moieties. As a result, the network structure is formed, which corresponds to the system gelation. The LCSTs of MC and HPMC are between 40 to 50 °C and 75 to 90 °C, respectively [26], but chemical or physical modification may result in higher desired values [56,59,60].

Cellulose ethers (CEs) are other bioadhesive polymers. Cellulose is a linear homopolymer polysaccharide consisting of D-anhydroglucopyranose units joined together by ß-1,4-glycosidic bonds [61]. Cellulose is insoluble in water due to extensive intramolecular hydrogen bonding. Thermogelation of cellulose derivatives varies with the degree and type of substitution. CEs are produced by etherification of the three hydroxyl groups of cellulose anhydroglucose components, which produce water-soluble derivatives. The results are the production of CEs such as MC, HPMC and CMC [62]. CMC is an ionic ether of cellulose and it is the major commercial derivative in which original H atoms of cellulose hydroxyl groups are replaced by carboxymethyl substituent [63]. However, MC and HPMC are used frequently. MC is a cellulose derivative that has been widely studied for biomedical applications. It is long-chain substituted cellulose consisting approximately 27–32% of the hydroxyl groups in the methyl ether form [61,63]. It has thermoreversible gelation properties in aqueous solutions and gells at temperatures between 60 and 80 °C and transforms again into solution at lower temperature [61,64,65]. HPMC is partly O-methylated and O-(2-hydroxypropylated) cellulose [63]. Methoxy residues of HPMC are responsible for the gelation, due to the increase in hydrophobic interactions and exclusion of water from heavily methoxylated regions of the polymer [61,66].

Sodium alginate (Alg-Na) is a naturally occurring mucoadhesive biopolymer (Figure 2d). This polysaccharide is obtained mainly from brown algae belonging to the Phaeophyceae and composed of α–L-guluronic acid and β-d-mannuronic acid residues. Alginate has several desirable properties, such as biodegradability, non-toxicity, biocompatibility, and low cost, making it a promising biopolymer for various applications in DDSs [67]. PVP is one of the most widely used polymers in medicine due to its solubility in water and its extremely low cytotoxicity (Figure 2b). Another advantage of using PVP is that it can be thermally crosslinked, resulting in outstanding thermal stability and high mechanical strength of the material [68,69]. Polyacrylates, such as carbopol and PCP, are the most effective in mucoadhesive action since they have a high molecular mass (Figure 2e). This feature offers a relatively high period of residence on the mucosa [49,70]. Carbopol has 58–68% of carboxylic groups which progressively undergo hydrogen bonding with sugar residues in oligosaccharide chains in the mucus membrane [71]. It results in the formation of a strengthened network between polymer and mucus membrane, so that carbopol having a high density of available hydrogen bonding groups is able to interact more strongly with mucin glycoproteins. It is speculated that enhanced mucoadhesive strength of the delivery system may lead to prolonged retention and increased absorption of drugs across mucosal tissues [71,72]. Carbopol, as a synthetic polymer, has been often used as a component of RDDSs. Due to its high viscosity, it may be used as the bioadhesive polymer to reinforce the gel strength of the P407/P188 thermosensitive hydrogel [53,73].

The addition of non-ionic surfactants is also often used in the manufacture of thermosensitive liquid suppositories. These compounds can act as wetting agents and can have a positive impact on the drugs release [74]. The addition of Tween (most commonly Tween 80) results in highly viscous gel formation, increased mucoadhesive strength and decreased gelation temperature and time. The mechanism by which Tween 80 influences the properties of the gel can be the reinforcement of the hydrogen bonding between the poloxamer combination in the gel matrix [75,76]. Non-ionic surfactants such as Tween 80 have been reported to be inert, resulting in no damage to mucous membranes [28]. In rectal administration, Tween 80 does not reveal any side effects.

In order to be used for treatment, the thermosensitive liquid suppository must have satisfactory rheological and mechanical properties, such as:gelation temperature: the temperature at which the liquid phase is transformed to the gel phase. The gelation temperature range that would be appropriate for rectal administration is 30–36.5 °C;viscosity: viscosity of the thermosensitive liquid suppository at 36.5 °C is known as gel strength; liquid suppository with optimal gel strength (10–50 s) will remain in the upper part of the rectum and will not leak out from the anus;gelation time and gel strength: thermosensitive liquid suppository with a relatively faster gelation time and optimal gel strength will remain in the upper part of the rectum and will not leak out from the anus. Gelation time means the time taken for the thermosensitive liquid suppository to achieve a viscosity of approximately 4000 mPa·s at 36.5 °C. Gelation time varies according to suppository composition, but is usually 2–8 min;mucoadhesive force: the force by which the thermosensitive liquid suppository binds to the mucous membranes of the rectal.

Reasonable rheological and mechanical properties can be obtained by using the different proportions of mucoadhesive polymers referred to above [38,77].

The most popular method of obtaining thermosensitive liquid suppository is cold method [78]. The mucoadhesive polymers can be used at different concentrations, as discussed later in this paper.

## 2. Application of Thermosensitive Liquid Suppositories as Innovative Systems for Delivering Various Drugs

One of the key advancements in modern pharmacology is the development of new drugs, new drug formulations or new DDSs. These solutions enable the delivery of active substances in a particular place, at the right time, using the most preferred form of administration, and with minimal side effects. The discovery of new synthetic drugs is, however, time consuming and expensive—thus, modern pharmaceutical science is generally focused on improving the pharmacokinetics of known drugs or developing innovative drug dosage forms. The production of thermosensitive liquid suppositories is one example. This dosage method modifies the pharmacokinetic properties of drugs and retains all the benefits of the rectal route. The available reports contain studies on thermosensitive liquid suppositories that deliver a range of drugs, e.g., analgesic, anticancer, antihypertensive, anaesthetic, etc.

### 2.1. Analgesic Drugs

Analgesic drugs, including nonsteroidal anti-inflammatory drugs (NSAIDs) are effective for the treatment of muscle pain, dysmenorrhea, arthritic conditions, pyrexia, gout, migraines and used as opioid-sparing agents in certain acute trauma cases. Most generally, NSAIDs are available as oral tablets or capsules. Topical NSAIDs are also available (e.g., topical solution, patch, gel). Specific NSAIDs can also be administrated parenterally (e.g., ibuprofen, ketorolac) [79]. Many drugs cannot be administrated in the above ways due to unfavourable properties, i.e., high first-pass effect, side effects, hydrophilic/lipophilic properties, etc.). An analgesic thermosensitive liquid suppository can contribute to the management of pain if the other route is not available.

Acetaminophen, commonly referred to paracetamol, was the first drug being used as a thermosensitive liquid suppository formulation. It is a drug with potent antipyretic and analgesic action, but very poor anti-inflammatory activity [80]. Choi et al. prepared the first acetaminophen-loaded liquid suppository in 1998 [81]. Novel in-situ gelling and mucoadhesive acetaminophen liquid suppositories were prepared with a gelation temperature of 30–36 °C with sufficient bioadhesive force and gel strength. P188 and P407 were applied to confer temperature-sensitive gelation properties. The solution P188 (15–20%) and P407 (15%) were found to be liquid at room temperature, but gelled at 30–36 °C. Bioadhesive polymers such as PVP, HPMC, hydroxypropyl cellulose (HPC), carbopol and PCP have been added to modulate the properties of acetaminophen liquid suppositories. However, they have had different effects on the physicochemical properties of this dosage type. Carbopol and PCP reduced the gelation temperature, whereas PVP, HPMC and HPC did not significantly affect it. At the same time, PCP and carbopol most significantly enhanced both bioadhesive force and gel strength. These thermosensitive liquid suppositories were inserted into the rectum of rats without difficulty and leakage, and retained in the proper place for at least 6 h [81]. Choi et al. continued their research in this field [82]. The authors produced thermosensitive liquid suppositories consisting of P407:P188:PCP:acetaminophen (15:19:0.8:2.5% *w/w*) and studied the drug release profile and its pharmacokinetics. They assumed that it was dependent on the components used. P188 demonstrates a slight impact on the release of acetaminophen from this dosage form. At the same time, PCP increased gel strength and bioadhesive force, therefore significantly delaying the release kinetics of acetaminophen. The release rates did not vary between the non-PCP suppository and 0.2% PCP-loaded suppositories. This parameter started to decrease as the concentration of mucoadhesive polymer increased by more than 0.4%. The analysis of the drug release mechanism showed that acetaminophen might be released by Fickian diffusion. The thermosensitive liquid suppository effectively formed gel in the rectum. It demonstrates a more sustained release profile than other suppositories and provided the most prolonged plasma levels of acetaminophen in vivo.

In addition, the thermosensitive liquid suppositories demonstrated a higher drug bioavailability than traditional dosage forms and did not do any damage to the rectal tissues. It remained stable for at least 6 months during storage [82]. Mucoadhesive acetaminophen liquid suppositories with sodium alginate have also been prepared and evaluated [83]. This formulation was composed of acetaminophen:P407:P188:sodium alginate (5:15:19:0–1.0% *w*/*w*. Satisfactory gel temperature, gel strength and bioadhesive force have been achieved for thermosensitive liquid suppository composed of [acetaminophen:P407:P188:sodium alginate (5:15:19:0.6%)], but this type had a similar release pattern to conventional suppositories. The parameters such as: biological half-life t_1/2_, mean residence time (MRT), the area under the drug concentration–time curve (AUC) and apparent elimination rate constant (K_el_) of acetaminophen from thermosensitive liquid suppositories were not significantly different from those of conventional form [83]. However, thermosensitive liquid suppositories were characterized by a higher maximum plasma drug concentration (C_max_) and lower time to achieve the maximum plasma concentration (T_max_) than conventional ones. In addition, acetaminophen liquid suppositories were simple to administer to the anus, more convenient for the patient and demonstrated quicker absorption of drugs in humans than conventional rectal dosage form [83].

Thermosensitive liquid suppositories have been established with the next analgesic drug, etodolac—an NSAID with analgesic properties [11,84]. Rectal etodolac–poloxamer gel systems composed of poloxamers (P188, P407) and bioadhesive polymers (HPMC, PVP, CM, HEC, carbopol) were developed and evaluated. The characteristics of the thermosensitive liquid suppositories differed according to the properties of mucoadhesive polymers. Modulation of the adhesive and physicochemical properties of P407 and P188 mixtures by mucoadhesive polymers showed a prolonged in-vitro drug release. The mucoadhesive polymers (HPMC, MC, HEC, and PVP) were examined at different concentrations ranging from 5–15%. As the concentration of these mucoadhesive polymers increased, the gelation temperature of all obtained materials decreased. At the same time, the addition of carbopol to the mucoadhesive polymer (1:1 ratio) showed higher gelation temperature. Moreover, all mucoadhesive polymers have improved gel strength relative to suppositories without them. The higher the concentration of the mucoadhesive polymer, the higher the gel strength of the poloxamer gels. A significant increase in mucoadhesive force was observed in all types: HEC 15%-containing liquid suppository displayed the highest mucoadhesive force. Furthermore, the release of etodolac was variously affected by mucoadhesive polymer concentrations: it was retarded by the addition of mucoadhesive polymers relative to thermosensitive liquid suppository without the mucoadhesive polymers. HEC and MC showed the highest retardation, followed by PVP and HPMC. Furthermore thermosensitive liquid suppository of etodolac did not cause any morphological damage to the rectal tissues [11]. Kim et al. developed a thermosensitive flurbiprofen liquid suppository ([flurbiprofen/P407/P188/sodium alginate/HP-β-CD (1.25/14/13/0.6/22%)]) composed of poloxamers and sodium alginate and additionally used also α, β, γ-cyclodextrin as well hydroxypropyl-β-cyclodextrin (HP-β-CD) to enhance the aqueous solubility of the drug [85,86]. The results of studies on this dosage form showed that HP-β-CD decreased the gelation temperature and reinforced the gel strength and bioadhesive force of liquid suppository, while flurbiprofen did the opposite. In addition, flurbiprofen-liquid suppositories with HP-β-CD displayed higher plasma level, C_max_ and AUC of flurbiprofen than those without HP-β-CD. After rectal administration, the AUC of flurbiprofen did not vary significantly from that of commercial Lipfen (flurbiprofen axetil-loaded emulsion on intravenous administration), after intravenous administration [86]. The investigation of the effect of bioadhesive polymers on the drug release and pharmacokinetic profiles were also the aim of another study. Ozgüney and coworkers developed thermosensitive and mucoadhesive liquid suppositories with ketoprofen (KP), a non-steroidal anti-inflammatory drug [31,87]. The material was prepared using KP, P407, P188 and various amounts of different mucoadhesive polymers, such as: PVP, CMC, C, HPMC. The release of KP was variously affected by the type and concentration of mucoadhesive polymers. *In vitro* release studies showed that carbopol 934 has a significant effect on the drug release rate among the mucoadhesive polymers. It affected the KP release rate from the concentration of 0.2% and the release rate decreased with increase in carbopol concentration. For the formulations prepared with HPMC as mucoadhesive polymer, the decrease of release rate was 4% up to the concentration of 0.6%. The decrease of release rate for the formulations containing CMC and PVP in the concentration of 1.6% was 8 and 5%, respectively. Additionally, identification of poloxamer gel localization in vivo showed that the suppositories remain in the rectum without leakage after administration [31]. The same author and other coworkers also investigated gelation temperature, viscosity and mechanical properties of thermosensitive and bioadhesive liquid suppositories containing KP [40]. He used bioadhesive polymers again to modify properties of this rectal dosage form. Bioadhesive thermosensitive liquid suppositories were produced by the cold method using P407, P188 and other bioadhesive polymers (poloxamers, CMC, HPMC and PVP) in various amounts. The study showed that in the presence of KP, the gelation temperature of the formulation P407/P188 (4/20% *w*/*w*) significantly decreased from 64.0 to 37.1 °C. It was found that the decrease of gelation temperature with addition of carbopol as bioadhesive polymer was higher than the decrease of gelation temperature with the addition of HPMC, CMC and PVP. The addition of bioadhesive polymers at higher concentrations lowered the gelation temperature and its decrease was the highest with the addition of carbopol 934. Furthermore, the gel structure of the thermosensitive liquid suppository composed of P407/P188/KP/C (4/20/2.5/0.8% *w*/*w*) showed the greatest hardness, compressibility, adhesiveness and viscosity [40].

Thomas et al. evaluated the effect of various mucoadhesive polymers (such as MC, HPMC, HEC, and CMC) on the physicochemical properties of thermosensitive liquid suppositories with lornoxicam [88], a highly potent NSAID [89]. Thermosensitive liquid suppositories were prepared by the cold method using a combination of poloxamers: P407 and P188. In this material, the gelation temperature decreased and the gel strength increased by increasing the concentration of P407, while increasing the concentration of P188 had the opposite effect. Importantly, the increasing concentration of poloxamers decreased the drug release rate. Among all the mucoadhesive polymers examined, HEC and CMC had the highest retardation in drug release, while HPMC had the least. The release retardation effect can be ranked as follows by a type of mucoadhesive polymer: CMC > HEC > MC > HPMC. Furthermore, the addition of mucoadhesive polymers reinforced mucoadhesive strength and gel strength. The average increase in gel strength observed with the mucoadhesive polymers used may be arranged in the following order: HPMC < MC < HEC < CMC, whereas the mucoadhesive polymers may be arranged according to their mucoadhesive force enhancement effect at the thermosensitive liquid suppositories concentration of 1.5% as MC > CMC > HPMC > HEC [89]. Al-Wiswasi and coworkers also studied the effect of mucoadhesive polymers (HPMC, PVP, sodium alginate) on the physicochemical properties of thermosensitive liquid suppositories with naproxen, a non-steroidal anti-inflammatory drug [90,91]. Gelling liquid suppositories using P188 (26–30%) as a suppository base with 10% naproxen have been prepared and evaluated in situ. Increasing the concentration of P188 from 26% to 30% was followed by a decrease in the gelation temperature. Moreover, the incorporation of 10% of the drug into P188 solutions at various concentrations reduced the gelation temperature. PVP, HPMC and sodium alginate have been used at concentrations ranging from (0.25–1%) to modulate properties, such as gelation temperature and gel strength. The effect of the additives on these properties was found to be depend on their nature and concentration. Increasing the concentration of each of the used mucoadhesive polymers from 0.25 to 1.0% resulted in a rise in gel strength and a gradual decrease in the gelation temperature of the naproxen-liquid suppositories. The optimal formulation was obtained by using a mixture of P188, sodium alginate, naproxen and distilled water (29, 0.5, 10 and 60.5%, respectively), with a gelation temperature of 33.6 °C ± 0.2 and a gel strength of 28 ± 2 s. The release of naproxen from this formulation was sustained for approximately 12 h and was faster compared to oral tablets and traditional solid suppository [90]. The main aim of Yuan et al. research was to establish thermosensitive and mucoadhesive rectal in situ gel with nimesulide (NM) by using mucoadhesive polymers, such as sodium alginate and HPMC [92]. NM is selective for inhibition of COX-2, has potent pyretolysis, analgesic and anti-inflammatory effects [93]. These materials were prepared by the addition of mucoadhesive polymers (0.5% *w*/*w*) to the formulations of thermosensitive gelling solution: P407 (18%) and NM (2.0%). The gelation temperature increased significantly with the addition of NM (2.0%), while the addition of mucoadhesive polymers played an opposite role in the gelation temperature. The addition of poly(ethylene glycol) (PEG) increased the gelation temperature and the drug release rate. Among all the formulations tested, P407/NM/sodium alginate/PEG 4000 (18/2.0/0.5/1.2% *w*/*w*) exhibited sufficient gelation temperature, reasonable drug release rate and rectal retention at the administration site. In addition, the initial serum concentrations, C_max_ and AUC of NM were substantially higher compared to the solid suppository [92].

Sodium tolmetin (TS) is a non-steroidal anti-inflammatory drug. As another NSAID, TS has side effects on the GIT following oral administration [94]. Poloxamer P407/P188-based thermosensitive TS-liquid suppositories were prepared by using mucoadhesive polymers such as MC [95]. The observations showed that the optimized TS-LS composed of P407/P188/MC (21/9/0.5% *w*/*w*) displayed gelling at rectum temperature ~32.90 °C, gel strength of 21.35 s and rectal retention of 24.25 × 102 dyne/cm^2^. Moreover, TS-LS did not cause any morphological damage to the rectal tissue [95]. Pharmacokinetic parameters of TS-LS showed a 4.6-fold improvement in bioavailability compared to Rhumtol^®^ capsules (commercial capsule with tolmetin sodium (TS) [95]. In relation to the drug release rate, Ramadan et al. also examined the safety of novel thermosensitive liquid suppository and drug hepatotoxicity [30]. Ketorolac tromethamine (KT) loaded mucoadhesive liquid suppositories were prepared and poloxamer mixture formed by 21% P407 and 9% P188 was used as a thermosensitive liquid suppository base. Unfortunately, KT may be hepatotoxic [96]. The results showed that the addition of KT increased the gelation temperature of the poloxamer gel and decreased the mucoadhesive force and gel strength. The KT release rate from the thermosensitive liquid suppository was significantly higher than that of the traditional one. Most notably, there was no hepatocellular damage after administering a thermosensitive liquid suppository to the anus as opposed to oral administration. Hepatotoxicity was not observed with this dosage form [30]. There are also articles describing preparation of mucoadhesive liquid suppositories with another analgesic drug diclofenac-nonsteroidal, anti-inflammatory drug with analgesic, anti-inflammatory and anti-pyretic properties [97]. The thermosensitive liquid suppository was developed using sodium chloride instead of bioadhesive polymers [98]. Liquid suppositories containing diclofenac could not be prepared using bioadhesive polymers because the drug was precipitated during their synthesis. Sodium chloride has been used to control the gel strength and bioadhesive force of thermosensitive liquid suppository. These rectal forms were composed of poloxamers (P407/P188:15/15% and 15/20%), sodium chloride (0–1%) and diclofenac sodium (0, 2.5%). The drug significantly increased the gelation temperature and weakened the bioadhesive force and gel strength, while sodium chloride had the opposite effect. Thermosensitive liquid suppositories of less than 1.0 wt.% sodium chloride have been inserted into the rectum without any difficulty or leakage. In addition, thermosensitive liquid suppository resulted in faster T_max_ of diclofenac sodium and significantly higher initial plasma concentrations than solid suppository [38]. Yong et al. also produced a liquid diclofenac suppository system by application of sodium chloride instead of bioadhesive polymers, and then studied its physicochemical properties (gel strength, gel temperature and bioadhesive strength) of various formulations composed of diclofenac sodium, poloxamers and sodium chloride [98]. They came to a conclusion similar to that of the group of scientists mentioned above. The mixture of poloxamers (P407:15% and P188:15–20%) was liquid at room temperature and gelled at physiological temperature. The analgesic diclofenac sodium significantly increased gelation temperature while decreasing gel strength and bioadhesive force, whereas sodium chloride had the opposite effect. Furthermore, thermosensitive liquid suppositories containing less than 1.0% sodium chloride were inserted into the rectum of rats for at least 6 h [98]. A liquid suppository was also created using a eutectic mixture [99]. Homogeneous eutectic mixtures with varying ratios of ibuprofen and menthol (0:10–10:0) were prepared. Ibuprofen-subsequent nonsteroidal anti-inflammatory agent [100] was used as a model agent for the preparation of ibuprofen-loaded liquid suppository using eutectic mixture with menthol and P188. It was assumed that the eutectic mixture and P188 would affect on the aqueous solubility of ibuprofen. In the absence of P188, the solubility of ibuprofen increased until the weight ratio of menthol to ibuprofen increased from 0:10 to 4:6, and then decreased when the ratio was above 4:6. This indicated that four parts of ibuprofen generated a eutectic mixture of six parts of menthol. In the presence of P188, solutions with the same menthol/ibuprofen ratio demonstrated a substantial increase in ibuprofen aqueous solubility. The gelling dosage form with a menthol/ibuprofen ratio of 1:9 and a concentration greater than 15% of P188 indicated the maximum solubility of ibuprofen (1.2 mg/mL). Ibuprofen increased the gelation temperature and weakened the bioadhesive force as well gel strength of the thermosensitive liquid suppositories, but menthol did the opposite. Ibuprofen-loaded liquid suppositories [P188/menthol/ibuprofen (15/0.25/2.5% *w*/*w*) has been easily administered to the anus and remain in the administered site without leakage. In addition, the C_max_, AUC, and initial plasma concentrations of ibuprofen were substantially higher than that of solid suppository [99]. Table 1 presents the thermosensitive liquid suppository components of analgesic drugs.

### 2.2. Anticancer Drugs

Antineoplastic agents can be administered through a variety of routes, including: intravenous, oral, intrathecal, intraventricular, intraperitoneal, intrapleural, intravesical, topical, subcutaneous and intramuscular [101]. Chemotherapy is also limited by toxicity. Intrarectal drug administration via a suppository may minimize this toxicity. The thermosensitive liquid suppositories can potentially be used as an anti-cancer drug carrier. This refers primarily to the treatment of lower gastrointestinal tract tumours.

5-Fluorouracil (5-FU) is an antimetabolite drug that is widely used for the treatment of cancer, particularly for colorectal cancer [102]. This drug was formulated into a thermosensitive liquid suppository, using P407 as a thermogelling agent for rectal administration in combination with pectin, which inhibited the drug’s solubility [46]. Additionally, the incorporation of carbopol 940 as a mucoadhesive polymer has altered the gelation temperature of P407. The selected thermosensitive liquid suppository formula P407:C940 (19:1 wt.%) showed a gelation temperature of 31.7 °C, gel strength of 75 s, and mucoadhesion strength of 2.476 × 103 dyne/cm^2^ [46]. Yeo et al. prepared and tested thermosensitive liquid suppositories with a different drug [103]. Docetaxel (DCT) is a wide spectrum anticancer drug and has been used as a model lipophilic drug with low water solubility and low oral bioavailability [78]. The main purpose of their work was to optimize the rheological properties of DCT thermosensitive liquid suppositories, prepared by a cold method. Important formulation criteria, including P407 and Tween 80, have been optimised to adjust the thermogelling and mucoadhesive properties for rectal administration. Tween 80 was chosen as a solubilizing agent as it formed a eutectic mixture with DCT. The rectal formulation remained as a liquid at room temperature and was converted into a gel at physiological temperature via a reverse gel phenomenon. All liquid suppositories with 10 wt.% Tween 80 underwent an evident sol-to-gel transition with a concentration of P407 ranging from 35 to 38%. The gelation temperature decreased with increasing in the concentration of P407. At higher concentrations of P407 (13 wt.%), the suppository gelled instantly, whereas the gelation time was longer with a decreased concentration of poloxamer. Furthermore, the viscosity increased with an increase in the concentration of P407. In addition, both formulation variables, P407 and Tween 80, had a major impact on the mucoadhesive force. The mucoadhesive force increased steadily with an increase in the concentration of P407. Similar findings were observed with an increase in the concentration of Tween 80. Based on these results, the thermosensitive liquid suppositories consisting of 0.25% DCT, 15% P188, 11% P407 and 10% Tween 80, were optimal and exhibited the following gel properties: mucoadhesive force 2.110 dyne/cm^2^, gel temperature 33.0 °C, and gelation time 7.5 min [103]. Thermosensitive liquid suppository has also been prepared with epirubicin (Epi), an amphiphilic anthracycline anticancer drug [104,105]. This formulation has been developed with temperature-sensitive P407 and pH-sensitive polyacrylic acid (PAA) at different ratios. In addition, the Epi-loaded solid suppository was also prepared to compare its properties with the non-drug liquid suppository. *In vitro* statistics revealed that Epi in P407 14%/ PAA, for both solid and thermosensitive liquid suppository, had substantial cytotoxicity, sustained drug release, long-term appropriate suppository base, strong bioadhesive force, and high accumulation of Epi in rat rectums [104]. P407 and PAA were found to be non-toxic, but increased chemosensitization of colon cancer cells to Epi. These liquid and solid suppositories were administered to the rectum of rats without difficulty and leakage and retained in the upper rectum for at least 12 h. This study has shown that Epi/P407/PAA inhibited colorectal tumor growth with reduced weight loss in solid and thermosensitive liquid suppository formulations [104]. Irinotecan is a water-soluble anticancer drug that is commonly used in the treatment of metastatic rectum and colon carcinomas as intravenous administration [106]. Due to very high plasma concentrations, irinotecan, which was administered for the treatment of rectal cancer, has significant side effects. Din and coworkers have produced a novel irinotecan-encapsulated double reverse thermosensitive nanocarrier system (DRTN) for rectal administration [107]. The DRTN was prepared by dispersing the thermosensitive irinotecan encapsulated solid lipid nanoparticles (SLNs) in the thermosensitive poloxamer (P407, P188) solution. The SLN is nanoparticulate drug carrier for controlled DDSs because it provides excellent drug release control and superior its encapsulation. This formulation consisted of 10% SLN dispersion, 15% P407, 17% P188, 4% Tween 80 and 54% water. The DRTN was readily applied to the anus, gelling quickly and strongly after rectal administration. The DRTN had a gel strength of almost 11.000 mPa/s, indicating that it developed a strong gel at 36.5 °C. There was no damage to the rat rectum and no loss of body weight. The DRTN resulted in sustained release and almost constant plasma concentrations of irinotecan at 1–3 h in rats, leading to an improvement in anticancer activity. The drug dissolution from DRTN was investigated in comparison to hydrogel. The DRTN provided a significantly lower drug dissolution rate, allowing the drug to be released for 1 h [107]. Oxaliplatin is a platinum-based chemotherapeutic agent [108]. In order to avoid the first-pass effect and reduce its toxicity, an in-situ-gelling and injectable P407–poly(acrylic acid) (P407–PAA) (P407:14%, PAA: 0.187%; 0.375%, 0.75%) thermosensitive liquid suppositories were developed. This dosage form was able to gel rapidly in physiological conditions and had sufficient gel strength and bioadhesive force [109]. Increasing the concentration of acrylic acid (AAc) from 0.187% to 0.75% and maintaining the other reaction conditions showed a substantial decrease in the gelation time from 222 to 170 s Moreover, the gel strength of the material increased from 18 to over 300 s when the concentration of AAc increased from 0.187% to 0.75%. In addition, the bioadhesive force of the formulation was also substantially improved with an increase in the concentration of AAc. Thermosensitive liquid suppositories were administered to the rectum of rabbits without difficulty and leakage for at least 6 h. Toxicity and cytotoxicity studies revealed that P407 and PAA were non-toxic and could inhibit colon cancer cells when oxaliplatin was added. C_max_ and AUC_0→12_ h oxaliplatin values were higher than oral drug solutions following rectal drug administration. It could be concluded that P407–PAA liquid suppository could avoid anticancer drugs undergoing the first-pass effect and reduce their toxicity [109]. Clotrimazole was first used against fungal infections, and a body of research was later developed to prove that this drug also has anticancer properties [110]. In order to establish a novel thermosensitive liquid suppository based on clotrimazole, the melting point of the different formulations consisting of P188 (50–100 wt.%) and propylene glycol (0–50 wt.%) was investigated [111]. The melting point of poloxamer solutions was significantly affected by P188. Precisely, the mixture of P188/propylene glycol (70%/30%) with a melting point of around 32 °C was solid at room temperature and immediately melted at physiological temperature. Furthermore, the ratio of P188/propylene glycol significantly affected the dissolution of the drug from the thermosensitive liquid suppository. Analysis of the dissolution mechanism revealed that the rate of dissolution of clotrimazole from poloxamer-based suppositories was time-independent. The clotrimazole-loaded rectal formulation with P188 and propylene glycol did not irritate or damage the rectal tissues of rats and resulted in improved antitumor activity in a dose-dependent manner in the mouse. In addition, rectal administration reduced hepatotoxicity relative to oral administration [111].

Table 2 showed the components of the thermosensitive liquid suppository with anticancer and antitumor drugs.

### 2.3. Antiemetic Drugs

The routes of administration of antiemetics are usually oral or intravenous, but patient compliance is often hindered by challenges associated with acute emesis or invasive parenteral administration [112]. In addition, vomiting is a classical problem of paediatric illness. The rectal route can be used as an alternative route for patients that complain about the difficulty of oral vomiting therapy.

Razek and coworkers developed an metoclopramide (MET) in situ gelling system as an efficient rectal route of administration [113]. MET is an effective antiemetic used to prevent various forms of emesis. It has a short half-life (about 4 h), which requires frequent administration [114]. MET-loaded liquid suppositories were prepared using P407 (20–25%) with mucoadhesive polymers such as HPMC, HEC or PVP at various concentrations (0.5%; 1.5%; 2.5%). Thermosensitive liquid suppository consisted of [P407-MET-HPMC 25%:2%:2.5% *w*/*w*] distinguished by the highest mucoadhesive force, viscosity and the slowest drug release. The gelation temperature of the prepared formulations ranged from 34.2 ± 0.121 °C. The formulation remains as a liquid at ambient room temperature and undergoes reversible thermal gelation at the body temperature. Furthermore, the mucoadhesive polymers could be arranged according to their mucoadhesive force-enhancing effect as follows: HPMC > HEC > PVP. HPMC, which has the highest molecular weight (∼86.000 g/mol), showed the highest mucoadhesive force. Additionally, with an increase in the concentration of poloxamer from 20 to 25%, the viscosity increased. Furthermore, mucoadhesive polymers presented an improved effect on viscosity. When the concentration of P407 was 20%, the increase in concentration of PVP from 0.5 to 2.5% resulted in a substantial increase in viscosity. A similar effect was observed with HPMC and HEC. The higher the viscosity, the slower the drug is released from the liquid suppository. All of the thermosensitive liquid suppository also demonstrated sustained MET release for 8 h. It was observed that the drug release was not only affected by the concentration of P407, but also by the type of mucoadhesive polymer used. Increasing in the P407 concentration from 20 to 25% at 2.5% of PVP contributed to a decrease in the amount of drug released. Similar behaviors were found at 2.5% of HEC and 2.5% of HPMC [114]. Ondansetron, a serotonin-3 receptor (5-HT3) antagonist, has been used for the prevention and treatment of nausea and vomiting [115]. The thermosensitive–mucoadhesive ondansetron liquid suppository (tmOLS) has been developed [116]. This formulation has been prepared using poloxamers (P407 and P188) and HPMC. The thermosensitive liquid suppository base consisted of P407, P188 and HPMC in the ratio of 18:20:0.8% *w*/*w* was in the liquid state at room temperature, and underwent gelation at body temperature. Thermosensitive liquid suppository composed of P407, P188 and HPMC in the ratio of 18:20:0.4% *w*/*w* and 18:20:0.8% *w*/*w* resulted in an appropriate gelation temperature for tmOLS in the desired range (35–37 °C). Furthermore, the addition of HPMC (0.4–1.5%) resulted in a decrease in the gelation temperature. Importantly, t_1/2_ of tmOLS was two-fold that of the oral solution. The area under the curve (AUC) was significantly higher in the tmOLS-treated group, indicating that the formulation bypassed the first-pass metabolism and that it was released slowly from the tmOLS [116]. Jadhav et al. also developed ondansetron-loaded liquid suppository [117]. The main aim of this study was to develop in-situ gelling mucoadhesive liquid suppositories of ondansetron by using mucoadhesive polymers: sodium alginate, MC and PVP. These formulations were prepared by adding mucoadhesive polymers (0.8%) to the thermally gelling liquid suppositories containing P407 (18%) and ondansetron (0.8%). Moreover, PEG 6000 (0.5–2.0% *w*/*w*) was used to adjust the gelation temperature. This value was marginally increased when the drug was added to the poloxamer solution, whereas the addition of mucoadhesive polymers was found to have a reverse effect. Although these polymers reinforced the gel strength and mucoadhesive force of the solutions, the addition of PEG 6000 increased the gelation temperature of the poloxamer solution. Thermosensitive liquid suppository consisting of MC required a higher concentration of PEG 6000 in order to modulate the gelation temperature in the range of 30–36 °C. Among the different formulations tested, the formulation [P407/ondansetron/sodium alginate/PEG 6000 (18/0.8/0.8/1.3% *w*/*w*)] demonstrated sufficient gelation temperature, mucoadhesive force, gel strength and suitable drug release profile. The higher mucoadhesive force of this rectal dosage form may result in the retention of the thermosensitive liquid suppository in the rectum. Thus, the first-pass metabolism of the drug would be avoided and, eventually, the bioavailability of ondansetron will be improved [117]. The components of the antiemetic thermosensitive liquid suppository are shown in Table 3.

### 2.4. Antihypertensive Drugs

Many antihypertensive drugs are generally administered orally. Exceptions include intravenous agents, which are typically used in emergency situations or when the oral route is not appropriate, as well as agents that can be administered via transdermal and sublingual routes [118]. However, there are some papers explaining the rectal administration of antihypertensive drugs, e.g., carvedilol and propranolol [119,120]. Importantly, this route of administration can improve the pharmacokinetics of antihypertensive drugs, but is unfortunately still not common.

Diltiazem hydrochloride is a calcium channel blocker that is commonly used for the treatment of hypertension, arrhythmia and angina pectoris [121,122]. Keny and coworkers have developed and tested theoreversible in situ gelling and mucoadhesive liquid suppositories with diltiazem to enhance systemic drug absorption [123]. These formulations were prepared by adding mucoadhesive polymers at a concentration of 0.5% or 1.0% (carbopol 974P, Polyox WSR-301 (polyethylene oxide), HPMC, PCP and PVP) to the formulation consisting of P407, P188 and diltiazem hydrochloride. The formulation containing P408/P188 in the ratio of 20/10% was selected as optimized for further study because it showed a gelation temperature in the range of 30–36 °C. The mean average decrease in the gelation temperature observed for all rectal gel formulations was found to be as follows: PCP > Carbopol > Polyox WSR-301 > HPMC > PVP. The effect of mucoadhesive polymers on gelation temperature was discovered to be dependent on their nature and concentration. Furthermore, increasing the concentration of any of the used bioadhesive polymers from 0.5 to 1.0% resulted in a decrease in gelation temperature. Furthermore, the addition of mucoadhesive polymers improved the gel strength of the poloxamer mixture in a concentration-dependent manner. Carbopol exhibited the highest gel strength. The average increase in gel strength observed with all the bioadhesives used may be arranged in the following order: PVP < Polyox WSR-301 < HPMC < PCP < C. In addition, the mucoadhesive polymers could be arranged according to their mucoadhesive force enhancing effect at 1.0% concentration of rectal gel: C > PCP > Polyox WSR-301 > HPMC > PVP. Increasing the concentration of mucoadhesive polymers also effected the release of diltiazem hydrochloride from the gelling formulation. Carbopol and PCP displayed the highest retardation of drug release relative to other mucoadhesive polymers investigated. The formulation with 1.0% of HPMC showed satisfactory results for the rheological parameters evaluated and produced a drug release of 80% at the end of 8 h [123].

Propranolol is a commonly used β-blocker that undergoes substantial hepatic first-pass elimination after oral administration, with systemic bioavailability of 15–20% [124]. Mucoadhesive liquid suppositories containing this drug were prepared by adding mucoadhesive polymers (0.6%) to thermally gelling suppositories containing P407 (15%), P188 (15%) and propranolol (2%) [125]. Carbopol, HPC, sodium alginate, PVP and PCP have been added and examined as mucoadhesive polymers. The characteristics of the suppository varied depending on the choice of the mucoadhesive polymer. Among the mucoadhesive polymers tested, sodium alginate and PCP demonstrated the largest mucoadhesive force and the smallest intrarectal migration resulting in the highest bioavailability of propranolol (84.7 and 82.3%, respectively). There was no relationship between bioavailability, gelation temperature and drug release profile. However, sodium alginate and PCP exhibited the highest retardation in release of propranolol, followed by carbopol, HPC and PVP. The gel strength of solution P407/P188 (15%/15%) increased as the concentration of sodium alginate, carbopol and PCP was increased, but was not affected by HPC and PVP [125]. Patil et al. formulated and characterized mucoadhesive liquid suppositories through the use of chitosan-based microspheres [126]. Candesartan cilexetil (angiotensin II receptor antagonist) was used as a model drug [127]. Chitosan-based microspheres were formulated using a single emulsification technique, and thermosensitive liquid suppositories were prepared by a cold preparation method using candesartan cilexetil-loaded chitosan microspheres. Percent of the drug encapsulation in the microspheres was found to be between 71.25% and 96.2%. The gelation temperature of all formulations ranged from 28 to 34.16 °C. Chitosan demonstrated good mucoadhesion properties in the rectum and prolonged drug release up to 12 h [126]. In Table 4 the components of thermosensitive liquid suppository with antihypertensive drugs are given.

### 2.5. Psychiatric Drugs

Oral use is the primary route of administration for psychiatric drugs. However, studies have discussed alternate routes, such as topical (patches), sublingual, intravenous or intramuscular administration [128,129]. Patients who do not tolerate any gastric stimulation can benefit from a medication that can be absorbed rectally (solid and liquid suppositories). This could be especially useful for patients who cannot tolerate sublingual treatment and for those who are unable to tolerate intramuscular or intravenous daily dosing [130].

Carbamazepine (CBZ) is indicated for epilepsy treatment. Unfortunately, after oral administration, the drug undergoes substantial hepatic first-pass elimination [131]. El-Kamel and coworkers prepared mucoadhesive CBZ-loaded thermosensitive liquid suppositories by applying carbopol to thermally gelling suppositories containing 20% of P407, 15% of P188 and 1% of MC [132]. All formulations contained 10% of the drug. The thermosensitive liquid suppository containing 20% of P407, 1% of MC and 0.5% of carbopol demonstrated the most appropriate rheological properties: gel temperature 30 ± 0.3 °C and bioadhesive force 16170 ± 50 dyne/cm^2^. The release mechanism study showed that CBZ was released from the thermosensitive liquid suppository through Fickian diffusion. The addition of 15% of P188 to the formulation containing 20% of P407 and 0.5% of carbopol marginally increased the release of the drug. MC was used instead of 15% P188 in the formulation containing 20% of P407 and 0.5% of carbopol to improve drug release characteristic from the gelling rectal formulation. Enhanced release of CBZ was observed as 50% of CBZ was released after 28 min. Furthermore, the findings showed that this thermosensitive liquid suppository had a relative bioavailability of 97.7% compared to produced oral CBZ suspension [132]. The goal of the next study was to develop and evaluate levosulpiride-loaded thermosensitive liquid suppository to improve its bioavailability [76]. Levosulpiride is used for the treatment of depression and psychiatric problems and has been used as a model drug in this research [133]. These rectal formulations were consisted of 15% of P407, 10–20% of P188, 2–5% of Tween 80 and 0.5–1.5% of the drug. P188 and Tween 80 decreased the gelation time and temperature as well improved mucoadhesive force and gel strength. Thermosensitive liquid suppository composed of [Levosulpiride/P188/P407/Tween 80 (1/15/17/3% *w/w*)] had a gelation temperature of around 30.7 °C. It remained liquid at 25 °C, but was transformed to a gel at 30–36.5 °C. Importantly, no leakage was observed in rats after rectal administration. The drug-loaded thermosensitive liquid suppository displayed significantly improved AUC values relative to levosulpiride suspension, resulting in a 7.1-fold improvement in bioavailability. In addition, the C_max_ of the rectal gelling formulation was slightly higher than that of the drug suspension. The higher AUC and C_max_ of the liquid suppository could be attributed to the improved bioavailability of levosulpiride [76]. The composition of psychiatric-loaded thermosensitive liquid suppositories is given in Table 5.

### 2.6. Insulin

Subcutaneous insulin injections (syringes, insulin pens and insulin pumps) are the most common route of insulin administration. There are, however, non-invasive modes of insulin administration. These include: oral, colonic, rectal, nasal, ocular, pulmonary, uterine and transdermal routes of administration. Unfortunately, at present, these alternative routes do not include clinically relevant solutions to the subcutaneous mode of administration [134].

He et al. investigated the effects of N-trimethyl chitosan chloride as an absorption enhancer to improve the properties of insulin lthermosensitive liquid suppository (with a quaternization degree of 42.77% (TMC40) and 63.03% (TMC60)) [135]. Insulin is a macromolecular peptide hormone produced by β-cells of the pancreatic islets and it is difficult to be administered by a gastrointestinal route because of inactivation by proteolytic digestion and poor absorption across mucosal membranes [135]. The scientists prepared thermosensitive liquid suppository composed of P407/P188/Insulin (15 /20 /0.36% *w/w*) with additional TMC40 (0.05; 0.1; 0.5; 1.0%) or TMC60 (0.05; 0.1; 0.5; 1.0%) or sodium salicylate (10%). Compared with sodium salicylate, which increased the gelation temperature and decreased the bioadhesive force and gel strength, both TMCs increased these three indices. Researchers concluded that TMC40 with a concentration higher than 0.1% and TMC60 with a concentration above 0.05% increased insulin bioavailability when compared to 10% of sodium salicylate. TMC was found to significantly increase the permeation of peptide analogues across intestinal epithelia. Furthermore, TMC significantly increased the intestinal absorption and bioavailability of peptide analogues with no evidence of epithelial damage or cytotoxicity [135]. Yun and coworkers have produced a thermosensitive insulin liquid suppository to improve its bioavailability [28]. The effects of insulin and sodium salicylate on the physicochemical properties of a liquid suppository composed of P407 (15%), P188 (20%) and PCP (0.2–0.6%) were examined. The findings showed that only thermosensitive insulin liquid suppository, composed of [insulin:P407:P188:PCP:sodium salicylate for 15:20:0.2:10% *w/w*] showed optimal physicochemical properties. Furthermore, the thermosensitive liquid suppository with 10–30% of sodium salicylate demonstrated lower plasma glucose levels in rats relative to thermosensitive liquid suppository without this compound. These findings have shown that insulin-liquid suppository with more than 10% sodium salicylate may be better absorbed in rats compared to sodium salicylate-free suppository due to enhanced absorption impact [28]. The ingredients of the rectal insulin dosage forms mentioned above are shown in Table 6.

### 2.7. Antiallergic Drugs

Allergic drugs may be administered in a range of ways, such as: oral, sublingual, cutaneous, subcutaneous, nasal, ocular, transdermal, pulmonary and rectal. The route of administration depends on the properties of the antiallergic drug and the symptoms of allergy that need to be treated [136].

Promethazine (PMZ) is an anti-allergic and anti-histamine drug [137]. Jaafar et al. developed and tested thermogelling mucoadhesive liquid suppository of promisehazine to improve its bioavailability and reduce its side effects [138]. P407 (12–20%) and P188 (4–12%) were used as thermogelling agents, as well HPMC (0.5–2%) as mucoadhesive polymer. The gelation temperature of the rectal formulation was substantially increased with the addition of 12.5% of PMZ; however, it decreased when HPMC was added. Furthermore, the mucoadhesive polymer improved the gel strength and the mucoadhesive force of the prepared solutions, but significantly lowered the drug release rate. Following formulations: P407/P188/HPMC (16/4/0.5%) and P407/P188/HPMC (16/4/0.75%) were selected for PMZ-loaded thermosensitive liquid suppository since they exhibited optimum physical properties of gelation temperature, gel strength, high mucoadhesive force and a moderate PMZ in-vitro release [138]. The components of antiallergic-loaded thermosensitive liquid suppository are given in Table 7.

### 2.8. Anaesthetic Drugs

The routes of administration of general anaesthetic agents include: intravenous, inhalational, intramuscular, oral (very rarely) and rectal (very rarely), while local anaesthesia may be caused by the injection, spray, gel, cream and plaster. Rectal anaesthetics are commonly used to alleviate the pain and itching of haemorrhoids (piles) and other conditions in the rectal area [141].

Lidocaine belongs to the class of local anaesthetics [142]. The goal of the [139] study was to implement the rapid onset of drug action to relieve pain and inflammation in the treatment of hemorrhoids—swollen and inflamed veins in the rectum and anus. A novel in situ gelling and mucoadhesive liquid lidocaine suppository has been developed. The rectal formulation consisted of lidocaine (2%), P407 (5–25%) and P188 (1–5%) and HPMC (0.5–1.0%), as well carbopol 940 (0.5–1.5%) as mucoadhesive polymers. This form was liquid at room temperature and gelled at 30–36 °C. Formulations containing carbopol 940 showed improved gel strength compared to the formulations containing HPMC. There has been an improvement in bioadhesive strength with an increase in polymers concentration. In addition, poloxamer gels without the addition of mucoadhesive polymers had less mucoadhesive force. Moreover, the drug release is delayed by the addition of mucoadhesive polymers. HPMC and carbopol containing formulations showed more than 80% drug released in 5 h [139]. All ingredients of anaesthetic thermosensitive liquid suppositories are shown in Table 7.

### 2.9. Antimalarial Drugs

Antimalarial drugs can be administered in a number of ways: oral (amodiaquine), intramuscular (artemether), intravenous (artesunate) [143]. There is also a scope for rectal administration of the antimalarial drug (quinine) [144].

Chloroquine phosphate (CP) still remains the drug of choice that is effective against infections caused by *Plasmodium Falciparum, Plasmodium Vivax, Plasmodium Malariae* and *Plasmodium Ovale* [145]. Rectal chloroquine-P407 gel systems composed of P407 (18–24%) and bioadhesive polymers (03–0.9%) such as PVP, carbopol 934P and PCP have been developed and evaluated [140]. The gelation temperature for the formulations varied between 32.4–36.5 °C, whereas mucoadhesive force was found to be in the range of 37.34–321.05 dynes/cm^2^. PCP and carbopol 934 showed higher mucoadhesive strength, retardation in drug release from P407 gels and significantly reduced the gelation temperature by about 60 °C [140]. The drug release was found to be matrix diffusion-controlled and the release mechanism was found to be Fickian diffusion. As the concentration of mucoadhesive polymer increased from 0.3% to 0.9%, the drug diffusion decreased from 100.49 to 98.52% for PVP, 96.52 to 90.45% for carbopol 934 and 95.20% to 89.06% for PCP [140]. The components of antimalarial thermosensitive liquid suppositories are given in Table 7.

## 3. Conclusions and Future Recommendations

Inadequate pharmacokinetic properties of many effective and tested drugs on the market have limited their application. Therefore, one of the key developments in modern pharmacology is to improve the pharmacokinetics of known drugs by creating new dosage forms and new DDSs. It is especially important for drugs with low bioavailability, significant first-pass effect, short half-life and multiple dosing frequency. The preparation of modern and innovative formulations enables the delivery of active substances at a particular site, at the right time and with minimal side effects. The thermosensitive liquid suppositories are an example of such a solution, especially that this dosage form could improve the pharmacokinetic properties of drugs with different mechanisms of action. The preparation of this innovative formulation is a new field of science. Unfortunately, as a number of papers have shown, it has not yet been thoroughly examined.

In this review, we summarized the recent developments in thermosensitive liquid suppositories as a delivery system for various drugs, such as analgesic, anticancer, antiemetic, antihypertensive, psychiatric, antiallergic, anaesthetic and antimalarial and insulin.

In general, the rectal route of administration is a safe alternative way to deliver drugs, particularly when the oral route is not feasible, due to unconsciousness, nausea or vomiting. The rectal route is also ideally suited for parenteral use in paediatric and geriatric patients. Conventionally solid suppositories are the most common dosage forms used for rectal drug administration but offer patients an alien feeling, discomfort, and result in refusal. The problems of traditional solid suppositories relevant to the above can be overcome by developing a thermosensitive liquid suppository. This drug form is simple to administer to the anus since it remains liquid at lower temperatures. In addition, it can act as a mucoadhesive to rectal tissue without leakage after dosage, does not cause any damage to mucosal layers and reduces the sensation of a foreign body compared to solid suppositories. It can also induce partial prevention of first-pass metabolism and rapid absorption of low molecular weight drugs. Other benefits of this approach include preventing overdosing, a constant and static rectal environment, minimizing exposure of the gastric mucosa to irritant drugs and shielding enzymatically unstable drugs such as insulin. Thermosensitive liquid suppositories can also enable the drug to circulate and cause not only a systemic but also a local effect, as with lidocaine. Unfortunately, in addition to the advantages mentioned above, this rectal formulation also has its disadvantages. First of all, the formulation of thermosensitive liquid suppositories is a new field of research. This could be correlated with high production costs, which is a very significant limitation and a problem. Insufficient knowledge on this subject can also lead to difficulties in the preparation of thermosensitive liquid suppositories. The selection of a temperature-sensitive polymer is an important element in the development of gelling suppositories since the biocompatibility and biodegradation of the rectal system depends on it. In addition, the choice of the proper form of mucoadhesive polymer and its concentration to obtain a thermosensitive liquid suppository with sufficient rheological properties can be challenging. Gelation time should also be monitored to prevent initial, rapid drug release. The amount of the drug released also depends on the exact composition of the thermosensitive liquid suppository. Another disadvantage is that not all drugs can be administered in this dosage form; e.g., rectal mucosa irritating drugs.

However, current study demonstrates that various drugs with different mechanism of actions can be delivered to the body in the form of thermosensitive liquid suppositories. Unfortunately, there are not sufficient research studies on drug release kinetics and mechanisms regarding composition of the liquid suppositories and the types of polymers used. Furthermore, the degradation and stability of thermosensitive liquid suppositories have not been demonstrated, including the degradation process, metabolism, metabolite fate and clearance time. As a result, this paper recommends further research in the aforementioned areas.

## Figures and Tables

**Figure 1 ijms-22-05500-f001:**
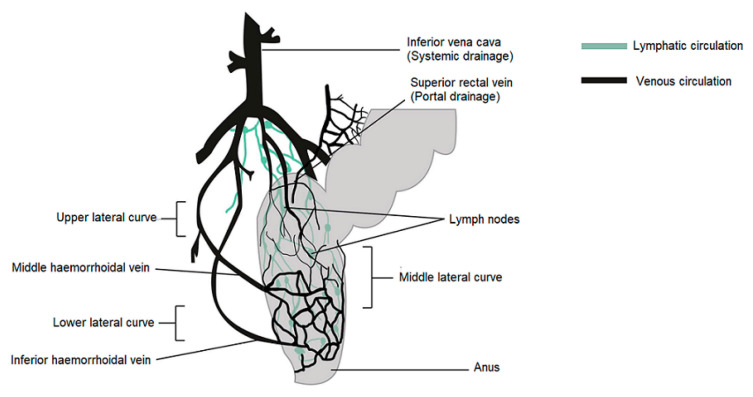
Venous and lymphatic drainage from the rectum.

**Figure 2 ijms-22-05500-f002:**
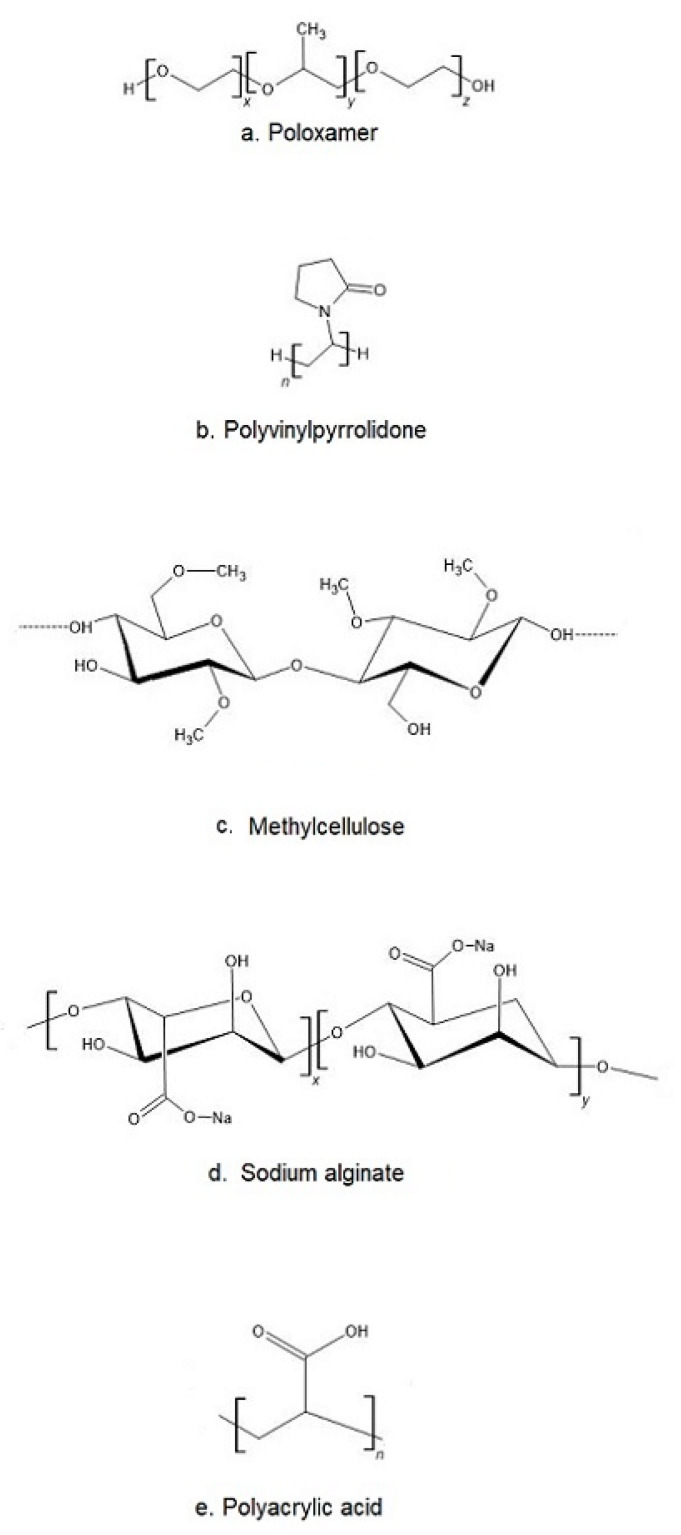
The chemical structure of exemplary polymers used in the production of thermosensitive liquid suppository: (**a**) poloxamer, (**b**) polivinylpyrrolidone, (**c**) methylcellulose, (**d**) sodium alginate, (**e**) polyacrylic acid.

**Table 1 ijms-22-05500-t001:** Thermosensitive liquid suppositories components of analgesic drugs and kinetics release/transport mechanism.

Drug	P407	P188	CARBOP.	HPMC	CMC	PVP	HPC	MC	HEC	PCP	Sodium Alginate	Tween	HP-β-CD	NaCl	Menthol	Kinetics/ Mechanism	Ref.
Etodolac 5%	15%	20%	5.0%	5.0% 10.0% 15.0%		5.0% 10.0% 15.0%			5.0% 10.0% 15.0%							Fickian diffusion	[11]
Ketorolac 10 mg	21%	9%						0.2% 0.6% 1.0%			0.2% 0.6% 1.0%					Fickian diffusion	[30]
Ketoprofen 2.5%	4%	20%	0.2% 0.4% 0.6% 0.8% 1.6%	0.2% 0.4% 0.6% 0.8% 1.6%	0.2% 0.4% 0.6% 0.8% 1.6%	0.2% 0.4% 0.6% 0.8% 1.6%										Higuchi model, non-Fickian diffusion	[31]
Diclofenac 2.5%	15%	15–20%												0–1%		Un-known	[38]
Ketoprofen 2.5%	44%	20%	0.2% 0.4% 0.6% 0.8%	0.2% 0.4% 0.6% 0.8%	0.2% 0.4% 0.6% 0.8%	0.2% 0.4% 0.6% 0.8%										Un-known	[40]
Acetaminophen 2.5%	12–15%	15–20%		1.0%		1.0%	1.0%									Un-known	[81]
Acetaminophen 2.5%	15%	15–20%								0.2% 0.4% 0.8 %						Fickian diffusion	[82]
Acetaminophen 5%	15%	19%									0.2% 0.4% 0.6% 0.8% 1.0%					Fickian diffusion	[83]
Flurbiprofen 1.25%	14%	13%									0.6%		20% 22% 25%			Un-known	[86]
Lornoxicam 0.16%	15–25%	7–20%		1.0% 1.5%	1.0% 1.5%			1.0% 1.5%	1.0% 1.5%							Fickian diffusion	[88]
Naproxen 10%		26–30%		0.25–1.0%		0.25–1.0%					0.25–1.0%					Un-known	[90]
Nimesulide 2%	18%	5–20%		0.5% 1.0% 2.0 %							0.5% 0.8% 1.0%					Anomalous (erosion-diffusion mechanism)	[92]
Tolmetin 5%	21%	9%		0.5% 1.0% 1.5%		0.5% 1.0% 1.5%		0.5% 1.0% 1.5%			0.6%	T20, T40, T80: 3%				Zero-order release kinetics, first-order release kinetics, Higuchi’s diffusion	[95]
Diclofenac 2.5%	15%	15–20%												0–1%		Un-known	[98]
Ibuprofen 2.5%		15%													0.25%	Un-known	[99]

**Table 2 ijms-22-05500-t002:** Thermosensitive liquid suppositories components of anticancer drugs and kinetics release/transport mechanism.

Drug	P407	P188	Carbopol	Tween 80	PAA	Propylene Glycol	Pektin	Kinetics/ Mechanism	Ref.
5-fluorouracil 1%	10–20%		1.0%				2.0%	Non-Fickian release	[46]
Docetaxel: 0.15% 0.20% 0.25%	10–13%	15%		5.0% 10.0% 15.0%				Un-known	[103]
Epirubicin 0.050%	14%				0.1875% 0.375% 0.75%			Un-known	[104]
Irinotekan 1%	15%	17%		4.0%				Un-known	[107]
Oxaliplatin 6.2 µM 12.3 µM 22.9 µM	14%				0.187% 0.375% 0.75%			Un-known	[109]
Clotrimazole 5%		50–100%				0–50%		Zero-order dissolution	[111]

**Table 3 ijms-22-05500-t003:** Thermosensitive liquid suppositories components of antiemetic drugs and kinetics release/transport mechanism.

Drug	P407	P188	PVP	HEC	HPMC	Sodium Alginate	MC	Kinetics/ Mechanism	Ref.
Metoclopramide 2%	20–25%		0.5% 1.5% 2.5%	0.5% 1.5% 2.5%	0.5% 1.5% 2.5%			Higuchi diffusion	[113]
Ondasetron 0.08%	18%	20%			0.4% 0.8% 1.0% 1.5%			Un-known	[116]
Ondasetron 0.8%	18%		0.8%			0.8%	0.8%	Fickian diffusion	[117]

**Table 4 ijms-22-05500-t004:** Thermosensitive liquid suppositories components of antihypertensive drugs and kinetics release/transport mechanism.

Drug	P407	P188	PVP	HPMC	HPC	Carbopol	Polyox Wsr-301	PCP	Chitosan	Kinetics/ Mechanism	Ref.
Candesartan 0.2%	1.8%								0.2–0.8%	Un-known	[126]
Diltiazem 0.018%	20%	10%	0.5–1.0%	0.5–1.0%		0.5–1.0%	0.5–1.0%	0.5–1.0%		Fickian diffusion (Higuchi model)	[123]
Propranolol 2%	15%	15%	0.6%		0.6%	0.6%		0.6%		First-order release kinetics	[125]

**Table 5 ijms-22-05500-t005:** Thermosensitive liquid suppositories components of psychiatric drugs and kinetics release/transport mechanism.

Drug	P407	P188	MC	Carbopol	Tween 80	Kinetics/ Mechanism	Ref.
Levosulpiride 0.5% 1.0% 1.5%	15%	10–20%			2–5%	Fickian diffusion	[76]
Carbamazepine 10%	20%	15%	1.0%	0.5%		Fickian diffusion	[132]

**Table 6 ijms-22-05500-t006:** Thermosensitive liquid suppositories components of insulin and kinetics release/transport mechanism.

Drug	P407	P188	TMC40	TMC60	Sodium Salicylate	PCP	Kinetics/ Mechanism	Ref.
Insulin 0.38%	15%	20%			10–30%	0.2–0.6%	Un-known	[28]
Insulin 0.36%	15%	20%	0.05–1.0%	0.05–1.0%	10%		Un-known	[135]

**Table 7 ijms-22-05500-t007:** Thermosensitive liquid suppositories components of antiallergic, anaesthetic and antimalarial drugs and kinetics release/transport mechanism.

Drug	P407	P188	HPMC	Carbopol	PVP	PCP	Kinetics/ Mechanism	Ref.
Promethazine 12.5%	12–20%	4–12%	0.5–2.0%				Zero-order, first-order, anomalous (non-Fickian), Fickian diffusion	[138]
Lidocaine 2%	5–25%	1–5%	0.5–1.0%	0.5–1.5%			Un-known	[139]
Chloroquine phospate 0.15%	18–24%			0.3% 0.6% 0.9%	0.3% 0.6% 0.9%	0.3% 0.6% 0.9%	Fickian diffusion	[140]

## Data Availability

Not applicable.

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
