# Peer review of "Achievements in Thermosensitive Gelling Systems for Rectal Administration"

_ijms, 2021, doi:10.3390/ijms22115500_

Round 1
Reviewer 1 Report
The manuscript (Recent achievements in thermosensitive gelling systems for rectal administration by Maria Bialik, Marzena Kuras, Marcin Sobczak, Ewa Oledzka) provides an overview on rectal drug delivery system together with a new concept on thermosensitive liquid suppositories. It is interesting well structured and informative review paper.
General comment:
Information in manuscript are within the focused area of Pharmaceutics. Therefore, the paper may be further highlighted if it is published in the journal from MDPI-Pharmaceutics.
Specific comments:
- L 12: Change to “drug dosage form”
- L 22: Change to “reveal further”
- L 349-352: Change to “The discovery of new synthetic drugs is however time consuming and expensive, thus modern pharmaceutical science is generally focused on improving the pharmacokinetics of known drugs or developing innovative drug dosage forms. The production of thermosensitive liquid suppositories is one of the example”
- L 927: Replace “high” with “multiple”.
- L940: Replace “popular” with “common”
- L953-954: Change to “First of all, the formulation of thermosensitive liquid suppositories is a new field of research.”
- L957: Change “Liquid, thermosensitive suppositories” with “thermosensitive liquid suppositories» (It has to be uniform in other places too).
- L966-970: However, current study demonstrates that various drugs with different mechanism of actions can be delivered to the body in the form of thermosensitive liquid Unfortunately, there are not sufficient research studies on drug release kinetics and mechanism, regarding to composition of the liquid suppositories and the types of polymers used.
Author Response
Dear Reviewer,
We would like to thank you for your insightful remarks. We thoroughly investigated all of the concerns raised and revised the manuscript in accordance with the recommendations. Please find attached the revised version of our manuscript, with the corrected text highlighted in yellow type.

Reviewer 2 Report
This review represents an analysis of the developments in thermosensitive liquid suppositories as drug delivery system. However, the references of the last 5 years are less of 10%. Then, I suggest to remove "Recent" from the title and in the text.
Author Response
Dear Reviewer,
We would like to thank you for your insightful comment. We revised the manuscript in accordance with the recommendation. Please find attached the revised version of our manuscript, with the corrected text highlighted in yellow type.
